# What's the Real: A Novel Design Philosophy for Robust AI-Synthesized Voice Detection

## ABSTRACT

Voice is one of the most widely used media for information transmission in human society. While high-quality synthetic voices are extensively utilized in various applications, they pose significant risks to content security and trust building. Numerous studies have concentrated on AI-synthesized voice detection to mitigate these risks, with many claiming to achieve promising performance. However, recent researches have demonstrated that existing fake voice detectors suffer from serious overfitting to speaker-irrelative features (SiFs) and cannot be used in real-world scenarios. In this paper, we analyze the limitations of existing fake voice detectors and propose a new design philosophy, guiding the detection model to prioritize learning human voice features rather than the difference between the human voice and the synthetic voice. Based on this philosophy, we propose a novel AI-synthesized voice detection framework named SiFSafer, which uses pre-trained speech representation models to enhance the learning of feature distribution in human voices and the adapter fine-tuning to optimize the performance. The evaluation shows that the average EERs of existing fake voice detectors in the ASVspoof datasets can exceed 20% if the SiFs like silence segments are removed, while SiFSafer achieves an EER of less than 8%, indicating that SiFSafer is robust to SiFs and strongly resistant to existing attacks.

## CCS CONCEPTS

• **Security and privacy** → **Intrusion/anomaly detection and malware mitigation**; *Social network security and privacy*; • **Information systems** → **Multimedia information systems**.

## KEYWORDS

AI-Synthesized Voice; DeepFake; AI-synthesized voice detection; ASVspoof

## 1 INTRODUCTION

Voice is one of the primary mediators of information transmission in human society, and it plays a significant role in digital systems for instant messaging, trusted authentication, etc. High-quality synthetic voices are now extensively utilized in various applications. However, the proliferation of voice clone technology also poses significant risks - it threatens content security and can potentially undermine trust-building processes. Most traditional voice

*ACM MM, 2024, Melbourne, Australia*
© 2024 Copyright held by the owner/author(s). Publication rights licensed to ACM.
ACM ISBN 978-x-xxxx-xxxx-x/YY/MM
https://doi.org/10.1145/nnnnnnn.nnnnnnn

synthesis approaches rely on old-style techniques such as splicing and editing, often resulting in a discernible smoothness that human listeners can distinguish. In recent years, the advancement of artificial intelligence (AI) has significantly enhanced the quality of synthetic voice [31–33, 36, 41, 44], making them increasingly difficult to distinguish by human ears.

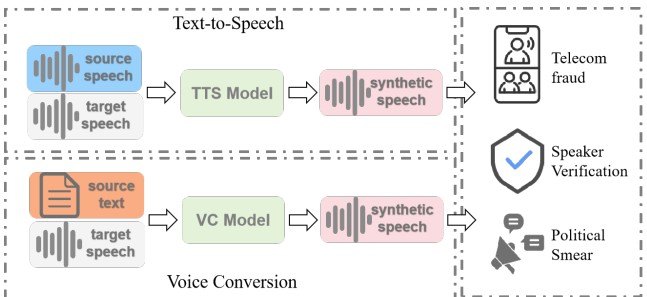

**Figure 1: Abuse of AI-synthesized voices. TTS and VC represent Text-to-Speech and Voice-Conversion respectively.**

Numerous incidents demonstrate that malicious actors have extensively exploited AI-synthesized voices to deceive authentication systems based on speaker verification, perpetrate telecom fraud, and even orchestrate political smear campaigns. In 2019, fraudsters utilized voice synthesis technology to convincingly mimic the voice of a CEO, successfully defrauding a substantial amount totaling over $243,000 [18]. Then, during the U.S. presidential election in New Hampshire in 2024, a significant number of voters received mysterious calls featuring a recording purportedly from U.S. President Biden, urging them not to vote in the state's primary. Subsequent investigation confirmed that the recording was generated using AI-synthesized voice [30]. These incidents underscore the abuse of voice synthesis technology for fraudulent and manipulative purposes, highlighting the urgent need for robust safeguards and countermeasures to mitigate such risks.

In the past few years, there has been a significant increase in research efforts focused on detecting fake voices. Early research primarily focused on extracting traditional speech features such as Mel-Frequency Cepstral Coefficients (MFCC) and spectrum to detect fake voices [3, 13]. In recent years, end-to-end (E2E)-based approaches have gained prominence. These methods utilize deep neural network (DNN) models to extract speech features and distinguish synthetic speeches directly. Most of them assert that their proposed methods are highly effective and perform excellently in their evaluation experiments. Some even report Equal-Error Rates (EER) lower than 1%, which is remarkable. However, several studies [8, 29, 46, 47] have pointed out that the noticeable difference in silence segments before and after human voice presented in the ASVspoof datasets (the most widely used datasets in AI-synthesized voice detection) could make all detectors trained and evaluated on

these datasets can easily detect spoof samples based on the silence segment alone. [9, 26, 43]. Recent research [15, 25] suggests that existing fake voice detectors exhibit serious robustness flaws and demonstrate significant fluctuations when applied to unfamiliar datasets. These studies indicate that the actual performance of existing fake voice detectors is not as excellent as portrayed in their experiments. They may pay excessive attention to the speaker-irrelative features (SiFs) that are not relative to the expression of voice, such as silence segments before and after the human voice, leading to a failure to capture the fundamental and essential differences between the synthetic voice and human voice. Consequently, most existing fake voice detectors perform poorly when facing silence-free voices or unfamiliar datasets, indicating that they are unsuitable for deployment in real-world scenarios.

The design objective of existing fake voice detectors is to discern the difference between synthetic voice and human speech. However, this often results in the detection model prioritizing the conspicuous differences between synthetic and genuine human voice in the training dataset. Consequently, the detector may become prone to overfitting, thereby disregarding the essential distinctions between synthetic voice and genuine human voice. In this paper, we propose a new philosophy for designing fake voice detectors, which aims to guide the detection model to learn what is "real" rather than the difference between the "real" and the "fake". We propose an AI-synthesized voice detection framework named SiFSafer, which employs the speech representation model as the upstream model to enhance the learning of genuine human voice feature distribution. We utilize a Bi-directional Long Short-Term Memory (BiLSTM) model to identify abnormal features that deviate from the genuine sample distribution to distinguish fake voices. We design a special upstream architecture comprising two identical speech representation models to optimize the detection performance. One model undergoes fine-tuning using adapter tuning, while the other remains an original model without fine-tuning, serving to mitigate model overfitting. Our main contributions are summarized as follows:

- We are the first to discuss the limitations of existing design philosophy, which guides models to learn the difference between synthetic and genuine human voices. And then, we propose that paying more attention to learning what is "real" is much more effective.
- We propose a new end-to-end fake voice detection framework named SiFSafer to counter the SiF-driven attacks. SiFSafer utilizes fine-tuned speech representation models and BiLSTM to detect fake voices effectively.
- We count a series of experiments to evaluate how SiFs influence existing detectors and the performance of SiFSafer. The results demonstrate SiFSafer's advantages over existing fake voice detectors.

The rest of this paper is structured as follows: Section 2 provides a brief overview of related technologies, including voice synthesis and AI-synthesized voice detection. Section 3 explains the details of SiFSafer, the AI-synthesized voice detection framework proposed in this paper. Section 4 outlines several experiments we designed to evaluate the performance of SiFSafer and existing detectors. Finally, in Section 5, we present our conclusions.

## 2 RELATED WORKS

### 2.1 Voice Synthesis

Voice synthesis can be divided into two categories: Text-to-Speech (TTS) and Voice Conversion (VC). TTS takes the text as input and generates the voice corresponding to the text. VC converts the timbre of the input voice to that of the target speaker.

The traditional TTS approaches generate new speech by concatenating pre-recorded segments, resulting in low-quality and poor flexibility. Most of the TTS approaches in recent years are based on Deep Neural Networks (DNN) to generate high-quality speeches. In 2016, DeepMind proposed WaveNet [39], which achieves powerful performance and gains widespread application. In 2017, Wang et al. [44] introduced Tacotron, an end-to-end TTS framework. Tacotron can directly map text to voice, but the performance is not as good as WaveNet's. Baidu proposed Deep Voice [4], which requires fewer parameters and achieves faster generation. In 2018, Shen et al. [32] introduced Tacotron 2, which improves the model structure and gets better performance compared with Tacontron. In 2021, Weiss et al. [45] proposed Wave-Tacotron, which extends the Tacotron model by incorporating a normalizing flow into the autoregressive decoder loop. Jaehyeon et al. [24] proposed VITS by combining three technologies: conditional variational autoencoder, normalizing flows, and Generative Adversarial Network (GAN). It gets ideal performance in speed and stability. In 2023, Wang et al. [41] proposed VALL-E, which is the first large-model-based TTS approach. It has excellent performance and supports zero-shot TTS.

Early VC approaches are usually implemented based on statistical transformation. The voice quality they generate is poor, and they require high-quality corpus data. In 2016, Hsu et al. [16] proposed a variational autoencoder-based VC approach that utilizes a display attribute vector to represent speaker information. In 2017, Kaneko et al. [20] proposed CycleGAN-VC, a GAN-based VC approach. In 2019, Kaneko et al. [21] proposed CycleGAN-VC2, which optimizes the adversarial loss and discriminator based on CycleGAN-VC.

### 2.2 AI-Synthesized Voice Detection

AI-synthesized voice detection approaches can be divided into four types: traditional features-based approaches, computer vision (CV)-based approaches, End-to-End (E2E)-based approaches, and other approaches.

Traditional features-based approaches transform the input voice to traditional voice feature representation, such as MFCC, and utilize the ML or DNN model to detect the AI-synthesized voice. In 2016, Tian et al. [37] compared the performance of six high-dimensional features by using a simple DNN model. In 2018, TODISCO et al. [38] transformed the voice to Linear Frequency Cepstral Coefficients (LFCC) and MFCC. Subsequently, they employed a Gaussian Mixture Model (GMM) for detection. Alzantot et al. [3] integrated several traditional features and proposed a detection scheme based on ResNet.

CV-based approaches are inspired by image recognition techniques. They first convert voice to image and then use deep-learning models designed for images to detect AI-synthesized voice. In 2019, Farid et al. [2] proposed the first CV-based approach which uses bispectral analysis to determine whether a sample is an AI-synthesized speech by Support Vector Machine (SVM). In 2021, Ballesteros et al.

proposed Deep4SNet [6]. It converts voice data into a histogram and uses a back-end model based on a Convolutional Neural Network (CNN) for classification.

E2E-based approaches have become the most popular approaches in AI-synthesized voice detection in recent years. They take raw voice data as input of the detection model and have no additional feature extraction. In 2020, Tak et al. proposed RawNet2 [35], which employs a sinc convolution layer to extract voice features and several residual blocks for further feature processing. In 2021, Tak et al. [34] proposed RawGAT which is based on a graph attention network. In 2023, Ding et al. [11] introduced a speaker attractor multi-center one-class learning approach to detect AI-synthesized voice. Guo et al. [14] employed a speech representation model WavLM for detection.

Some AI-synthesized voice detection approaches employ relatively rare technologies. In 2020, Wang et al. [42] proposed Deep-Sonar, which uses neuronal activity in an AI-based implementation of a speaker recognition system as a feature. In 2022, Blue et al. [7] used techniques from the field of articulatory phonetics for AI-synthesized voice detection.

## 3 METHODOLOGY

### 3.1 Motivation

Existing studies on AI-synthesized voice detection have reported ideal performance in their experiments, suggesting that accurate detection is achievable. However, recent researches [22, 25, 27] argues that existing fake voice detectors tend to overfit the SiFs, such as background noise, silence segments before and after human voice, etc. Researchers [15, 22, 25] have presented several novel SiF-based attacks to help AI-synthesized voices bypass detection, further emphasizing the limitations of existing detectors. These findings indicate that current fake voice detectors struggle to capture the essential differences between synthetic and human voices. Therefore, the challenge of AI-synthesized voice detection remains significant and far from resolved. The ultimate objective of AI-synthesized voice detection is to accurately distinguish all synthetic speech, regardless of the specific synthetic system used to generate it. However, there are two core difficulties in achieving this goal:

(1) **SiFs overfitting:** Since there is a significant distinction between data collection and processing, certain SiFs between synthetic and human voices can have noticeable differences. Fake voice detectors are prone to overfit SiFs, which may seriously affect the usability and robustness of the detection in the real world.

(2) **Dataset limitation:** Voice synthesis technology encompasses various methods and solutions. Synthetic voices generated by different voice synthesis systems may exhibit distinct feature distributions. Consequently, it is challenging for the datasets used in detection model training to encompass all types of synthetic voices. Fake voice detectors pay more attention to the difference between human voices and several specific types of synthetic voices, leading to diminished performance when encountering unfamiliar types of synthetic voices.

The design of existing detectors concentrates on capturing the disparity between synthetic and human voices. It leads the detection model to explore prominently distinct features, making it prone to overfitting SiFs or excessively emphasizing specific spoof features. To address the challenges above, we introduce a new philosophy for designing AI-synthesized voice detectors in this paper. Our core idea is to encourage the detection model to prioritize learning the feature distribution of human voice. We aim to identify synthetic voices by assessing whether the input data feature distribution aligns with human voices. This paper presents an AI-synthesized voice detection framework named SiFSafer based on this philosophy. SiFSafer utilizes a self-supervised learning speech representation model to reinforce understanding of human voice feature distribution. Additionally, we employ a BiLSTM-based network to identify outlier points in the distribution that deviate from human voice patterns.

### 3.2 Overview of SiFSafer Framework

We present the overview of the SiFSafer Framework in Figure 2. SiFSafer accepts raw voice data as input. Initially, we employ two self-supervised speech representation models: the original model released by the authors and a fine-tuning model, in which the latter is adapted based on the original model during our training phase. These models serve as the upstream feature extractors to generate the representation map of the input. The output of the upstream feature extractor consists of the outputs from each transformer encoder layer.

To enhance the feature representation capabilities of the representation map, we utilize a layer selection and fusion operation, which selects the output from specific layers and merges them into a unified representation. Following the fusion of the output using a linear layer to adjust dimensions, our approach employs a BiLSTM-based architecture to capture feature representations across frame levels comprehensively. Subsequently, SiFSafer integrates the outputs from the corresponding BiLSTM layer and maps it to the final output through a final linear layer.

### 3.3 Data Pre-Processing

The dataset is the basis of deep learning model training. Several studies [29, 46] have explored a notable duration disparity in silence segments before and after human voice between spoof and genuine samples within the ASVspoof datasets, which are extensively utilized in AI-synthesized voice detection.

As depicted in Figure 3, bonafide samples and spoof samples generated via voice conversion-based algorithms exhibit pronounced silence segments before and after human voice. Conversely, spoof samples generated by TTS algorithms lack such silence segments. It will mislead detection models to utilize the difference to distinguish the spoof samples and fail to learn other essential information. This work [15] demonstrates that existing fake voice detectors can be readily deceived by introducing silence segments to spoof samples. To enhance the robustness of SiFSafer, we remove the silence segments at the input stage to prevent potential overfitting.

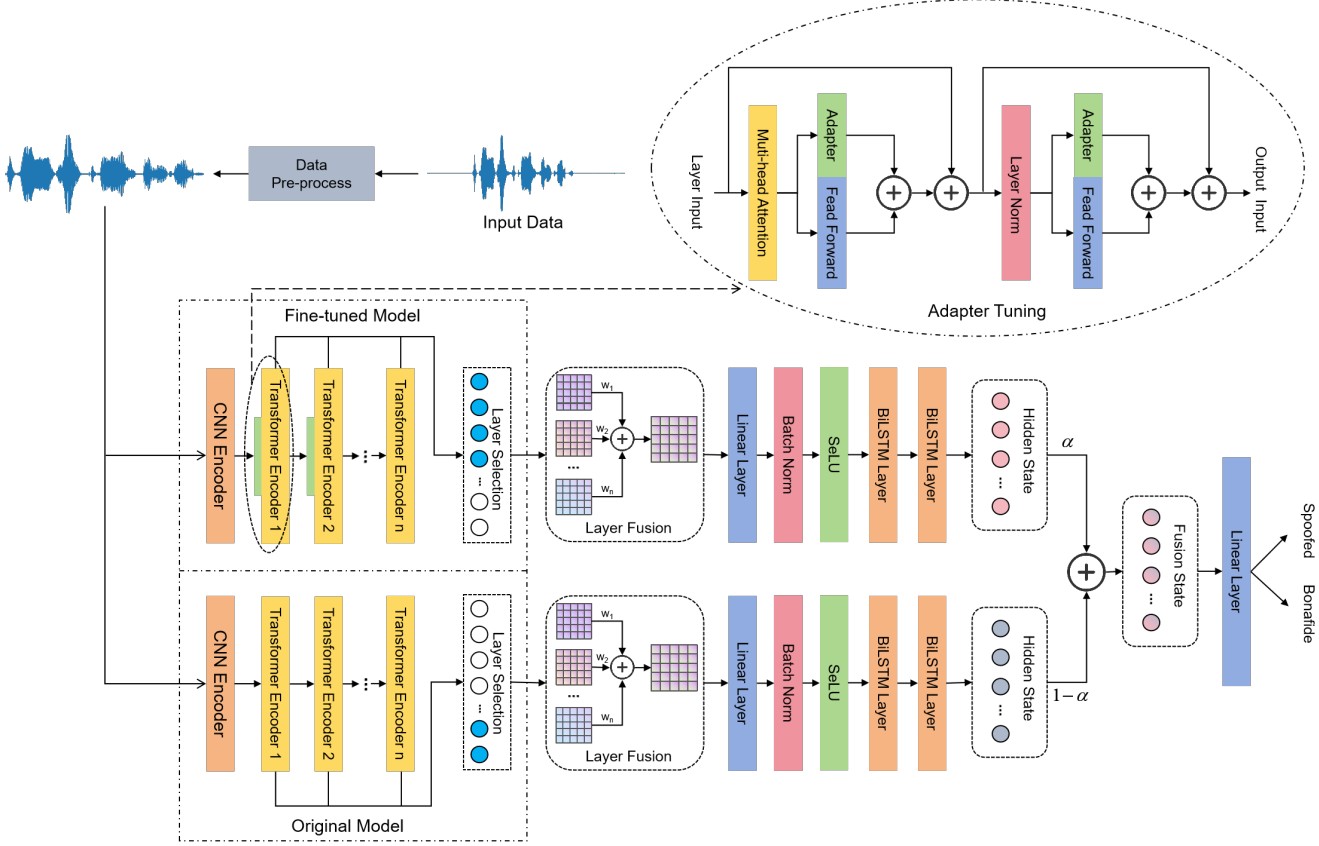

Figure 2: Pipeline of SiFSafer.

## 3.4 Upstream Selection and Fine-Tuning

The primary objective of SiFSafer is to comprehensively capture the human voice's feature distribution and identify synthetic speech by detecting anomalies that deviate from this distribution. However, the limited size of existing datasets in fake voice detection poses a challenge in achieving this goal. To address this limitation, we leverage one of the most widely adopted pre-trained speech representation models, wav2vec2.0 XLS-R [5], as the upstream feature extractor. This model comprises a convolutional neural network (CNN)-based feature extractor followed by a transformer-based [10, 40] feature encoder, trained on a massive dataset spanning 436k hours of speech data across 128 languages. The extensive training data enables the model to grasp genuine speech features effectively.

To further optimize the performance of this upstream model, we insert the LoRA adapter [17] into the transformer encoder layers shown in Figure 2 to fine-tune the model in the training stage. To avoid overfitting, we just insert the adapter into the initial layers, which mainly focus on the shallow features of voices, and freeze the parameter weights after several epochs of training. Simultaneously, to preserve the genuine speech information learned during pre-training to the fullest extent possible, SiFSafer utilizes an original model without additional fine-tuning to extract features parallel with the fine-tuned model.

## 3.5 Layer Selection and Fusion

The upstream feature extractor comprises multiple transformer encoder layers. In this architecture, the initial layers of the transformer encoder are responsible for extracting shallow feature information, while the subsequent layers delve deeper into the feature extraction process. Each layer takes the output of the preceding layer as input, enabling the model to progressively capture more abstract and intricate features of the input voice data.

The difference in feature distribution between synthetic and human voices may appear in feature information at all levels. So, paying reasonable attention to different levels of feature information is essential. The wav2vec2.0 model employed in this study encompasses a 24-layer transformer encoder. For the fine-tuned model, we select the output of the first 12 layers of transformer encoders, where the adapters are inserted, to extract fine-tuned shallow features. For the original layers, we select the output of the last 12 layers. This selection includes the information from the initial layers of the original model as well, as each layer takes the output of the preceding layer as input. This design choice enables SiFSafer to obtain a more comprehensive range of feature information spanning from shallow to deeper levels.

In the fusion process, the features selected from the output of the fine-tuned and original models are merged into two-dimensional

feature maps respectively. For each one, let $[O_1, O_2, ..., O_n]$ represent the feature information of n layers selected. We define a set of learnable weight values $[w_1, w_2, ...w_n]$. The fusion process is as follows:

$$FusionMap = w_1O_1 + w_2O_2 + ... + w_nO_n \quad (1)$$

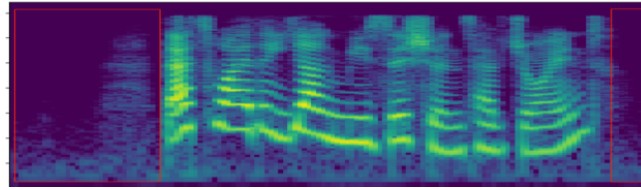

**(a) Bonafide samples**

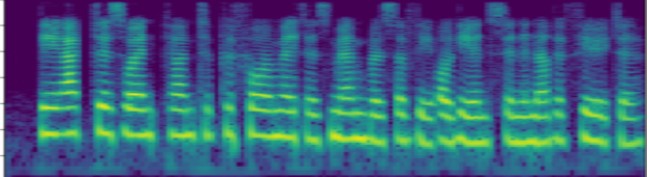

**(b) TTS based spoof samples**

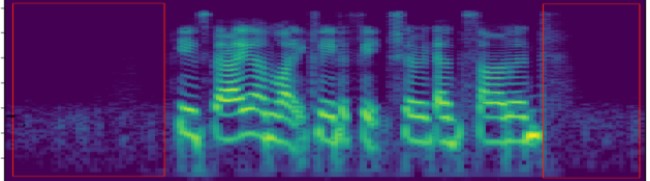

**(c) Voice conversion-based spoof samples**

**Figure 3: Silence segments difference in ASVspoof2019.**

## 3.6 Downstream Model

The downstream model captures the abnormal feature distribution from the fusion map, which is not similar to human voices. The fusion map can be viewed as a frame feature sequence. Initially, we utilize a linear layer to adjust the dimensions of features within each frame. Additionally, we apply the SeLU activation function to introduce non-linearity, thereby facilitating further fine-tuning of the upstream model. Subsequently, SiFSafer employs a two-layer BiLSTM network. This network architecture is adept at capturing cross-frame level feature clues by analyzing the sequential dependencies within the frame feature sequence.

To ensure the ability to capture feature clues at different levels, the fine-tuned and original models are equipped with independent downstream models, each possessing the same structure. This design choice enables SiFSafer to maximize the extraction of valuable insights from both the fine-tuned shallow features and the deeper features of the original model. Both the two outputs of BiLSTM layers are combined through a weighted sum using a learnable weight factor $\alpha$. Finally, a linear layer is applied to map the combined result to the final output.

## 4 EVALUATION

Our evaluation aims to answer the following research questions:

- **RQ1:** Can existing fake voice detectors work properly without silence segments before and after human voice?
- **RQ2:** Whether SiFSafer can get better performance than existing fake voice detectors?
- **RQ3:** Whether retraining the existing fake voice detectors without silence segments changes the result of the performance comparison in RQ2?
- **RQ4:** Whether SiFSafer is robust to the attacks based on SiFs?

## 4.1 Implementations and Datasets

This section will briefly introduce the implementation details of SiFSafer, baseline detector selection, and dataset selection in the experiments. Specific information is as follows:

**1) SiFSafer:** The technical details of SiFSafer are shown in the Section 3. The optimizer utilized during training is "Adam," with an initial learning rate set to 0.000001. For the fine-tuned model, we integrate the LoRA adapter into the initial 12 layers. To mitigate overfitting, we freeze the parameters of all LoRA adapters after 10 epochs.

**2) Baseline detectors**: We selected six recent fake voice detectors published in top conferences or related challenges: RawNet2 [35], RawGAT-ST [34], AASIST [19], MTLISSD [28], SAMO [11], FastAudio [12], and two pre-train model-based approaches by Piotr et al. [23], which serves as our baseline systems. RawNet2 serves as one of the baseline systems in ASVspoof2021 [1], and other systems have also demonstrated ideal performance on ASVspoof datasets. We acquired their implementations from open-source repositories provided by the respective authors. Initially, we intended to include DeepSonar, a neural network feature-based approach, as one of the baseline systems. However, due to the unavailability of open-source code from the authors and our unsuccessful attempts to reproduce their results, which significantly differed from their claims, we decided not to include it in our final selection.

**Table 1: Statistics of ASVspoof datasets. The ASVspoof2021 dataset contains only evaluation subsets.**

| Dataset | Speakers | Spoofing Algorithms | Conditions | Samples | |
|---|---|---|---|---|---|
| | | | | Spoof | Bonafide |
| 19 LA train | 20 | 6 | 1 | 22,800 | 2,580 |
| 19 LA dev | 20 | 6 | 1 | 22,296 | 2,548 |
| 19 LA eval | 67 | 13 | 1 | 63,882 | 7,335 |
| 21 LA eval | 67 | 13 | 7 | 163,114 | 18,452 |
| 21 DF eval | 93 | 110 | 9 | 589,212 | 22,617 |

**3) Datasets:** In this paper, we chose the ASVspoof2019 LA dataset as the training dataset for SiFSafer and all the baseline detectors. We selected three datasets for evaluation: ASVspoof2019 LA, ASVspoof2021 LA, and ASVspoof2021 DF. The specifics of these datasets are detailed in Table 1.

## 4.2 Existing Fake Voice Detectors Evaluation (RQ1)

In this evaluation, we mainly answer RQ1, *i.e.*, whether existing fake voice detectors work properly without silence segments before and after human voice. To evaluate the performance, we remove the silence segments before and after the human voice in all evaluation datasets using Sound eXchange (SoX), a cross-platform audio editing software. For baseline detectors, we use the respective authors' default parameters of training to retrain the models with the ASVspoof2019 LA dataset without any pre-processing because some of them do not provide pre-train parameters' weight.

**Table 2: The performance of baseline detectors in ASVspoof2019 LA evaluation set. The raw set is the original set and the silence set is the set after removing silence segemets.**

| Detector | Raw Set | | Silence Set | |
|---|---|---|---|---|
| | EER | min-tDCF | EER | min-tDCF |
| RawNet2 | 5.49% | 0.1453 | 28.96% | 0.6764 |
| RawGAT-ST | 1.51% | 0.0431 | 32.28% | 0.8411 |
| AASIST | 1.77% | 0.0510 | 30.96% | 0.7651 |
| Fastaudio | 1.79% | 0.0481 | 32.56% | 0.7879 |
| SAMO | 1.21% | 0.0307 | 23.03% | 0.5216 |
| MTLISSD | 2.58% | 0.0633 | 37.65% | 0.8247 |
| Whisper-lcnn | 20.84% | 0.6239 | 22.84% | 0.6733 |
| Whisper-specrnet | 23.03% | 0.6521 | 24.34% | 0.6832 |

The evaluation results are shown in Table 2. For the two pre-train model-based approaches by Piotr et al., we encountered unexpected performance results. Despite using the default parameters provided by the authors to retrain the models, we observed severe overfitting issues. Evidently, the performance significantly deteriorates in the dataset after removing silence segments, with all systems exhibiting a surprising degradation in performance. They all achieve EERs higher than 20%, indicating that they are unsuitable for real-world applications. It demonstrates that the silence segments are crucial in distinguishing fake voices for these systems. These systems are highly vulnerable to synthetic voice with no silence segments.

**Table 3: The performance of baseline detectors in ASVspoof2019 LA development set.**

| Detector | Raw Set | | Silence Set | |
|---|---|---|---|---|
| | EER | min-tDCF | EER | min-tDCF |
| RawNet2 | 1.02% | 0.0343 | 10.28% | 0.2837 |
| RawGAT-ST | 0.87% | 0.0273 | 17.39% | 0.4849 |
| AASIST | 1.18% | 0.0389 | 14.40% | 0.3765 |
| Fastaudio | 0.00% | 0.0000 | 12.76% | 0.3246 |
| SAMO | 2.20% | 0.0660 | 7.69% | 0.2355 |
| MTLISSD | 0.16% | 0.0048 | 27.31% | 0.5754 |
| Whisper-lcnn | 11.46% | 0.3624 | 14.36% | 0.4409 |
| Whisper-specrnet | 14.59% | 0.4408 | 16.01% | 0.4555 |

To further analyze the factors contributing to the performance degradation of the baseline systems, we evaluate their performance on the ASVspoof2019 LA development set. The results are presented in Table 3. The trend of performance change after removing silence segments is consistent with that observed on the evaluation set, but the performance degradation is relatively minor. We speculate that the better performance is because the spoofing algorithms present in the development set are consistent with those encountered during training. This further corroborates our hypothesis that the features specific to certain synthetic methods in the training set may contribute to overfitting.

## 4.3 SiFSafer Evaluation (RQ2)

In this evaluation, we mainly answer RQ2, *i.e.*, whether SiFSafer can perform better than existing fake voice detectors. We evaluate the SiFSafer and all the baseline detectors in ASVspoof2019 LA, ASVspoof2021 LA, and ASVspoof2021 DF datasets. All of the samples of the datasets are processed to remove the silence segments because detection based on silence segments has significant risks in real-world scenarios. To further analyze the performance of SiF-Safer, we also add a comparative detector comprising a wav2vec2.0 model identical to SiFSafer without fine-tuning and a downstream model proposed in this paper.

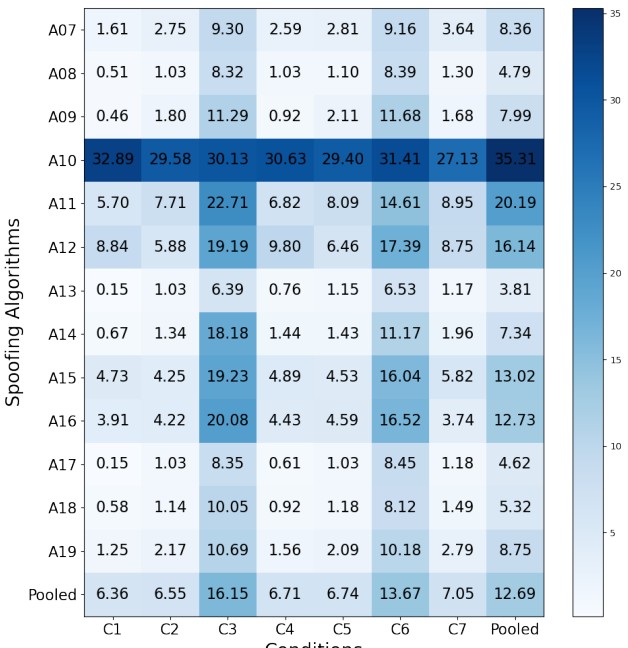

**Figure 4: The detail of performance of SiFSafer on ASVspoof2021 LA evaluation set. The data in each cell represents the corresponding EER.**

The comparison result is shown in Table 4. It is obvious that the performance of SiFSafer significantly outperforms that of the baselines. The average EER of most baseline detectors exceeds 20%, rendering them entirely unsuitable for real-world scenarios. In contrast, SiFSafer achieves an average EER of 7.41%. While the performance may not be optimal, it indicates that SiFSafer is suitable for real-world scenarios. The performance of the comparative

**Table 4: The performance comparison between SiFSafer and baseline detectors.**

| Group | Detector | 19 LA Set | | 21 LA Set | | 21 DF Set | Average EER |
|---|---|---|---|---|---|---|---|
| | | EER | min-tDCF | EER | min-tDCF | EER | |
| Baseline Detector | RawNet2 | 28.96% | 0.6764 | 34.39% | 0.8863 | 32.23% | 31.83% |
| | RawGAT-ST | 32.28% | 0.8411 | 49.64% | 0.9988 | 46.72% | 48.95% |
| | AASIST | 30.96% | 0.7651 | 32.15% | 0.8839 | 26.63% | 28.94% |
| | Fastaudio | 32.56% | 0.7879 | 28.93% | 0.7943 | 28.13% | 29.87% |
| | SAMO | 23.03% | 0.5216 | 36.48% | 0.9917 | 34.78% | 31.43% |
| | MTLISSD | 37.65% | 0.8247 | 43.75% | 0.9999 | 47.91% | 43.10% |
| | Whisper-lcnn | 22.84% | 0.6733 | 23.30% | 0.7539 | 11.07% | 19.07% |
| | Whisper-specrnet | 24.34% | 0.6832 | 24.41% | 0.7507 | 12.23% | 20.33% |
| Ours | Wav2vec2.0+downstream | 13.19% | 0.3282 | 15.03% | 0.5132 | 9.58% | 12.6% |
| | SiFSafer | **4.85%** | **0.1228** | **12.69%** | **0.5023** | **4.70%** | **7.41%** |

detector is also better than all of the baselines, suggesting that our design approach of utilizing a speech representation model to enhance the learning of genuine speech is effective. The performance gap between the comparative detector and SiFSafer highlights that fine-tuning one of two identical upstream models can significantly enhance the detection performance of the detector.

It should be noted that the performance of SiFSafer in the 2021 LA dataset is noticeably inferior compared to other datasets. To identify the reason, we conducted further analysis on the evaluation data of SiFSafer in the ASVspoof2021 LA dataset, as depicted in Figure 4. The performance in synthetic voices generated by A10 appears to be the poorest across all conditions. We hypothesize that this could be attributed to the high quality of this method, as similar performance is observed in other systems as well. Another clear trend is that the performance in C3 and C6 is worse than other conditions.

Condition C3 entails transmission over a PSTN system in Spain, where codec conditions are uncontrollable and unknown [26]. Condition C6, on the other hand, features the lowest bitrate among all conditions, with both conditions utilizing an 8kHz sampling rate. We speculate that the complex transmission environment and poor voice quality may result in the loss of detailed features, thereby making the overall distribution of synthetic voice closer to genuine human voice. The result also means that SiFSafer performs better in most conditions than the average on the ASVspoof2021 LA dataset.

## 4.4 The Baseline Detectors Evaluation after Retraining (RQ3)

In this evaluation, we mainly answer RQ3, *i.e.*, whether retraining the existing fake voice detectors without silence segments changes the result of the performance comparison in RQ2. We removed the silence segments of the ASVspoof2019 LA dataset and proceeded to retrain all baseline detectors using this modified dataset and the same training parameters as Section 4.2. This step was taken to mitigate the risk of the model overfitting to the silence segments. Performance evaluation was conducted using the same datasets described in Section 4.3.

The evaluation results in ASVspoof2019 are shown in Table 5. The performance of the majority of baseline detectors has demonstrated significant improvement. This suggests that by removing the silence segments from the training dataset, the model is directed

to pay more attention to the distinctions between synthetic and human speech, which might not have been effectively learned when utilizing the unprocessed dataset. However, it should be noted that the model's performance remains unsatisfactory despite the retraining efforts. The SiFSafer still has a very significant performance advantage.

**Table 5: The performance of re-trained baseline detectors in ASVspoof2019 LA evaluation set. The raw model represents the models trained in Section 4.2 and The silence model represents the models trained in this evaluation.**

| Detector | Raw Model | | Silence Model | | EER diff |
|---|---|---|---|---|---|
| | EER | min-tDCF | EER | min-tDCF | |
| RawNet2 | 28.86% | 0.7236 | 23.64% | 0.5608 | ↓ 18.09% |
| RawGAT-ST | 32.28% | 0.8411 | 22.50% | 0.4671 | ↓ 30.30% |
| AASIST | 28.06% | 0.7748 | 24.50% | 0.5119 | ↓ 12.69% |
| Fastaudio | 32.56% | 0.7879 | 19.69% | 0.4497 | ↓ 39.53% |
| SAMO | 23.03% | 0.5216 | 18.49% | 0.3926 | ↓ 19.71% |
| MTLISSD | 37.65% | 0.8247 | 23.43% | 0.5916 | ↓ 37.69% |
| Whisper-lcnn | 22.84% | 0.6733 | 22.52% | 0.6642 | ↓ 1.40% |
| Whisper-specrnet | 24.34% | 0.6832 | 23.94% | 0.6735 | ↓ 1.64% |
| SiFSafer | - | - | 4.85% | 0.1228 | - |

The performance evaluation of the baseline detectors after retraining using the ASVspoof2021 dataset is shown in Table 6. The baseline detectors continue to exhibit poor performance, with SiFSafer maintaining a clear lead in terms of performance. It is worth noting that Whisper-lcnn and Whisper-specrnet perform relatively well in the ASVspoof2021 DF dataset compared to those in other datasets. We infer the reason is that the pre-trained model Whisper has promising human voice modeling capabilities to help the detectors capture the features that do not belong to human speech in multiple spoof algorithms. The ASVspoof2021 DF dataset contains a more significant number of samples and spoof algorithms compared to the ASVspoof2019 LA dataset. The learning capability of Whisper regarding human voice features enables the detectors to perform better on the ASVspoof2021 DF dataset, even though it is overfitted on the ASVspoof 2019 LA dataset. Once again, the results indicate that solely focusing on the differences disparities

between human and synthetic voice within a specific dataset will significantly constrain the capabilities of fake voice detectors.

**Table 6: The performance of re-trained models in ASVspoof2021 dataset.**

| Detector | LA Set | | DF Set | Average |
|---|---|---|---|---|
| | EER | min-tDCF | EER | EER |
| RawNet2 | 29.38% | 0.7941 | 33.95% | 31.66% |
| RawGAT-ST | 27.96% | 0.8057 | 27.84% | 27.90% |
| AASIST | 30.90% | 0.8262 | 28.49% | 29.70% |
| Fastaudio | 20.73% | 0.5289 | 21.31% | 21.02% |
| SAMO | 29.83% | 0.7979 | 42.43% | 36.13% |
| MTLISSD | 32.29% | 0.9128 | 35.37% | 33.83% |
| Whisper-lcnn | 21.22% | 0.7202 | 6.82% | 14.02% |
| Whisper-specrnet | 22.66% | 0.7224 | 10.02% | 16.34% |
| SiFSafer | **12.69%** | **0.5023** | **4.70%** | **8.70%** |

## 4.5 Adversarial Attack Resistance Capability Evaluation (RQ4)

In this evaluation, we mainly answer RQ4, *i.e.*, whether SiFSafer is robust to the attacks based on SiFs. We selected SiFDetectCracker [15], the state-of-the-art attack approach against fake voice detectors that utilize SiFs, to evaluate the robustness of SiFSafer. We use the open-source code provided by the authors and default initial attack parameters for the implementation. The same method for sample selection, as described in their paper, is employed to obtain the test samples. Detectors of varying types exhibit different sensitivities to different SiFs. Therefore, we selected detectors employing diverse technical approaches to compare with SiFSafer. In this experiment, we selected a CV-based detector, Deep4SNet [6], along with two end-to-end based detectors, RawNet2 and AASIST, as comparison detectors. RawNet2 is one of the baselines used in the ASVspoof2019 challenge, and AASIST exhibits the best performance in the experiments reported in the papers, corresponding to all baseline detectors.

**Table 7: The result of robust evaluation.**

| Detector | Success Rate | Average Number of Iteration |
|---|---|---|
| Deep4SNet | 88.50% | 14.6 |
| RawNet2 | 80.40% | 13.9 |
| AASIST | 57.28% | 57.8 |
| SiFSafer | **0%** | **100** |

The evaluation result is shown in Table 7. The attack success rate against all comparison detectors is higher than 50%, whereas it is 0% against SiFSafer. The average number of iterations for the comparison detectors is lower than 50, indicating that the attack can be completed with a small number of queries. The results indicate that SiFSafer exhibits excellent robustness against SiFs and can effectively defend against the most existing advanced attacks.

## 5 CONCLUSION

In this paper, we analyze the limitations of existing fake voice detection designs and propose a new design philosophy aimed at directing the detection to focus more on learning the feature distribution of human speech. Based on this philosophy, we introduce a new fake voice detection framework named SiFSafer, which utilizes a pre-trained speech representation model to enhance the learning of human speech feature distribution. The evaluation demonstrates that existing fake voice detectors perform poorly when SiFs like silence segments are removed, while SiFSafer outperforms these existing detectors. The adversarial attack resistance capability evaluation also shows that SiFSafer exhibits excellent robustness against SiFs. SiFSafer's absolute performance still has significant room for improvement, particularly in handling low-quality speech. In future work, we aim to optimize the performance of SiFSafer further to address these limitations.

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
