# OpenReview forum: "What's the Real: A Novel Design Philosophy for Robust AI-Synthesized Voice Detection"
_acmmm.org/ACMMM/2024/Conference — MM2024 Poster_

### Official Review · Reviewer_szMa · 2024-05-04

**Rating:** 3
**Confidence:** 3

**Summary:**

This paper analyzes the limitations of existing fake voice detectors. It proposes a novel AI-synthesized voice detection framework named SiFSafer, which uses pre-trained speech representation models to enhance the learning of feature distribution in human voices and the adapter fine-tuning to optimize the performance. The motivation behind the paper is commendable, focusing more on detecting speech content than the silence segments. However, the comparisons made in the experimental section are unfair, so I am inclined to reject this paper.

**Strengths:**

This paper proposes a new philosophy for designing fake voice detectors, which aims to guide the detection model to learn what is "real" rather than the difference between the "real" and the "fake." Besides, it proposes a new end-to-end fake voice detection framework named SiFSafer to counter the SiF-driven attacks. SiFSafer utilizes finetuned speech representation models and BiLSTM to detect fake voices effectively.

**Limitations:**

1. The paper claims to be the first to discuss the limitations of existing philosophy in the contributions but states in the abstract that previous work overfits speaker-irrelevant features. I think this is a significant contradiction.
2. I believe the experiments related to RQ2 are unfair. Your overall comparison should not be based on a dataset with all silence segments removed. Instead, the experimental results for RQ2 should be derived from the original dataset. Furthermore, your results lack the performance metrics of your method on the original dataset.
3. Section 4.5 employs adversarial examples which use the SiFs to attack. Logically, this attack targets the silences and background noise. I am puzzled by these results because your model does not specifically defend against it. Are you comparing your model's performance on data where silence has been removed against a baseline on data that includes silence segments? This may explain the discrepancies in the experimental outcomes.
4. There is considerable redundancy in the data presented in Tables 2, 4, and 5. I suggest the authors need to rethink and rewrite the experimental section to address this issue.

**Suitability:**

2

---

### Official Review · Reviewer_V6dB · 2024-05-14

**Rating:** 3
**Confidence:** 4

**Summary:**

This manuscript analyzes the limitations of existing AI synthetic speech detectors and points out that they are prone to overfitting speaker irrelevant features (SiFs) in real-world scenarios, such as the silent segments before and after the human voice.
The author proposes a new design philosophy that guides detection models to prioritize learning human speech features rather than differences between human voice and synthetic speech. Evaluation on the ASVspoof dataset shows that SiFSafer exhibits robustness to SiFs and strong resistance to existing attacks, with an average equal error rate (EER) of less than 8% compared to existing detectors in the absence of SiFs (such as silent segments).

**Strengths:**

1: The perspective of analyzing speaker-irrelated features is interesting.

2: The SiFSafer framework exhibits better performance and robustness compared to existing detectors after removing irrelevant features of specific speakers (such as silent segments), indicating that it can work more reliably in real-world applications.

**Limitations:**

1: I have some concerns about the fairness of the experiment, as removing silent segments from the voice itself is a forgery process, which may undermine the continuity of the signal and lead to misclassification by the model. If the author can clarify or prove it, I will modify my rating.

2: In the evaluation of adversarial sample defense performance, the author only compared it with SiFDetectCracker and lacked comparisons with traditional adversarial attacks such as MIFGSM、Malafide、VMI-FGSM, or universal attack[1].

3: The author lacks comparison with some advanced works, such as [2], which are based on self supervised large models and have better performance on the in-the-wild dataset.

[1]Xie Y, Shi C, Li Z, et al. Real-time, universal, and robust adversarial attacks against speaker recognition systems[C]//ICASSP 2020-2020 IEEE international conference on acoustics, speech and signal processing (ICASSP). IEEE, 2020: 1738-1742.

[2]Yang Y, Qin H, Zhou H, et al. A robust audio deepfake detection system via multi-view feature[C]//ICASSP 2024-2024 IEEE International Conference on Acoustics, Speech and Signal Processing (ICASSP). IEEE, 2024: 13131-13135.

**Suitability:**

2

---

### Official Review · Reviewer_nemW · 2024-05-23

**Rating:** 4
**Confidence:** 4

**Summary:**

The paper presents a novel framework, SiFSafer, aimed at addressing the limitations of existing AI-synthesized voice detectors by focusing on learning human voice features rather than differences between human and synthetic voices. The author presents novel insights into the detection of AI-synthesized voice attacks, specifically targeting the robustness of this detection. The main contribution of this paper lies in the identification of previously overlooked aspects, such as the presence of silence segments in spoof samples created by voice conversion algorithms, which could mislead detection models. The authors have conducted extensive experiments and provided a detailed analysis of the results, demonstrating SiFSafer's robustness and superiority over existing detectors.

**Strengths:**

The paper introduces a new design philosophy that prioritizes learning genuine human voice features, which is a significant departure from existing methods that tend to focus on the differences between human and synthetic voices. The introduction of SiFSafer, which leverages pre-trained speech representation models and adapter tuning, is an innovative approach that shows promise in enhancing robustness against speaker-irrelative features (SiFs). The robustness against SiFs and resistance to adversarial attacks is a strong point.

**Limitations:**

1. In the experimental design, the choice of comparison models mostly relies on RawNet front-end or manual features, which are significantly influenced by silence segments. The Whisper front-end also does not perform exceptionally. This suggests that the selected comparison models might not represent the SOTA methods, potentially inflating the effectiveness.
2. The manuscript could benefit from a deeper analysis of why certain spoofing algorithms, like A10, are more challenging for SiFSafer.
3. There is a lack of discussion on the explainability and interpretability of SiFSafer's decisions.
4. There is a concern that the model may still be prone to overfitting, particularly given that it is trained on datasets that may not fully represent the diversity of real-world voice data.

**Suitability:**

2

---

### Official Review · Reviewer_ypZQ · 2024-05-30

**Rating:** 5
**Confidence:** 3

**Summary:**

Thank you to all the authors for their efforts and for the opportunity to review the manuscript. In this paper, the authors first demonstrate that existing synthetic speech detection methods which perform well on ASVspoof datasets, show a decline in performance when speaker irrelevant features (SiFs), for example, silence regions are removed during evaluation. This is because they tend to overfit on SiFs during training instead of learning unique features of genuine human speech. A method called SiFSafer is proposed which is demonstrated to not overfit on SiFs, and shows good performance on ASVspoof datasets. Finally, it is shown that re-training existing baselines without silence segments does not restore their performance to the level with silence segments, proving the effectiveness of SiFSafer approach.

**Strengths:**

1. The abstract and the introduction of the paper are well-written and clearly describe the background and motivation behind the problem statement. A minor comment - at least one example of SiFs should be included in the abstract, for example, silence regions. Without this, just from the abstract, it is unclear what kind of features are speaker-irrelative.
2. Removal of silence segments at the input stage to prevent overfitting
3. Combination of original and fine-tuned model considered during training
4. Experiments in the paper are organized well in the form of research questions. Exhaustive experiments support the hypothesis and demonstrate the effectiveness of the approach.

**Limitations:**

1. Section 2.2 needs some additions. In CV-based approaches, besides using the word “image”, spectrograms and mel-spectrograms should also be briefly introduced, since they are the common image representations of audio.
2. There are some inconsistencies in Figure 1. Input for TTS Model should be text, which is shown as input for VC Model. Please correct/clarify why this is the case. Also, Figure 1 has not been referenced in text.
3. Reasons for selection of “wav2vec2.0 XLS-R” as the upstream model:  Were other architectures considered and compared to XLS-R? Is there any experimental evidence or theory behind the choice? This is important because XLS-R is computationally expensive with hundreds of million training parameters.
4. What are the values of alpha and 1-alpha. Is there an ablation which supports the choice of these values? It is important to include these hyperparameter choices in the manuscript to ensure reproducibility and also ensure the fact that neither of original or fine-tuned model dominates the other model.
5. Is there a reason behind selecting only 3 other baselines for comparison in Table 7 (Adversarial Attack Resistance Capability Evaluation)? How do other methods like Whisper-lcnn react to these kinds of adversarial attacks?
6. How does this method perform on the datasets which evaluate generalization performance, for example, In-the-Wild dataset?

There are minor typos in the paper which could be removed by another round of proof-reading:

1. Line 118 - The full stop before “[9, 26, 43]” is not required.
2. Line 193 - Tacotron needs to be spelt correctly.
3. Line 259 and 260 - “recent researches argues” should be “recent researches argue”.
4. ASVspoof is written as ASVSpoof2021 in Line 551.

Overall, the paper presents an interesting approach to prevent overfitting on SiFs. The experiments provided in the paper are thorough with some inconsistencies as described above. A proof-read is recommended to remove typos.

**Suitability:**

3

---

### Meta-Review · Area_Chair_6sTv · 2024-07-04

**Recommendation:** Accept (Poster)
**Confidence:** 4

**Metareview:**

This paper analyzes the limitations of existing AI synthetic speech detectors, highlighting their tendency to overfit speaker-irrelevant features (SiFs) in real-world scenarios, such as silent segments before and after human voice recordings. The reviewers' ratings are controversial, with half leaning toward the negative side and the other half toward the positive side. The positive reviewers acknowledge that this paper presents an interesting approach to prevent overfitting on SiFs and commend the thorough experiments provided. Conversely, the concerns raised by the negative reviewers seem somewhat inappropriate. For instance, they requested comparisons with a paper that was officially published after the submission deadline of this work.